# Pan-Cancer Profiling of Intron Retention and Its Clinical Significance in Diagnosis and Prognosis

**DOI:** 10.3390/cancers15235689

**Published:** 2023-12-01

**Authors:** Leihuan Huang, Xin Zeng, Haijing Ma, Yu Yang, Yoshie Akimoto, Gang Wei, Ting Ni

**Affiliations:** 1State Key Laboratory of Genetic Engineering, Collaborative Innovation Center of Genetics and Development, Human Phenome Institute, School of Life Sciences, Fudan University, Shanghai 200438, China; 18110700099@fudan.edu.cn (L.H.); 20210700037@fudan.edu.cn (X.Z.);; 2Iskra Industry Co., Ltd., Tokyo 103-0027, Japan

**Keywords:** intron retention, RNA-seq, cancer, diagnosis, prognosis

## Abstract

**Simple Summary:**

Alternative splicing generates transcripts that can promote tumorigenesis, but intron retention (IR), a form of alternative splicing, remains understudied. We investigated the role of IR in tumorigenesis and its potential clinical applications at the pan-cancer level. We identified IR events from The Cancer Genome Atlas (TCGA) that differ between tumor and normal samples, as well as IR events associated with patient survival. These IR events may modulate the expression of host genes, including those that play a role in cancer biology. For example, we experimentally validated a novel IR event within the *STN1* 5′UTR. The excellent performance of IR-based diagnostic and prognostic models and their biological importance in cancer suggest that IR is a potential biomarker and drug target.

**Abstract:**

Alternative splicing can produce transcripts that affect cancer development and thus shows potential for cancer diagnosis and treatment. However, intron retention (IR), a type of alternative splicing, has been studied less in cancer biology research. Here, we generated a pan-cancer IR landscape for more than 10,000 samples across 33 cancer types from The Cancer Genome Atlas (TCGA). We characterized differentially retained introns between tumor and normal samples and identified retained introns associated with survival. We discovered 988 differentially retained introns in 14 cancers, some of which demonstrated diagnostic potential in multiple cancer types. We also inferred a large number of prognosis-related introns in 33 cancer types, and the associated genes included well-known cancer hallmarks such as angiogenesis, metastasis, and DNA mutations. Notably, we discovered a novel intron retention inside the 5′UTR of *STN1* that is associated with the survival of lung cancer patients. The retained intron reduces translation efficiency by producing upstream open reading frames (uORFs) and thereby inhibits colony formation and cell migration of lung cancer cells. Besides, the IR-based prognostic model achieved good stratification in certain cancers, as illustrated in acute myeloid leukemia. Taken together, we performed a comprehensive IR survey at a pan-cancer level, and the results implied that IR has the potential to be diagnostic and prognostic cancer biomarkers, as well as new drug targets.

## 1. Introduction

Alternative splicing (AS) is a way for eukaryotes to produce multiple transcripts from a single gene by changing the composition of exons during RNA processing [1]. Intron retention (IR) is a type of AS and affects about 80% of human genes [2]. IR transcripts tend to be degraded by nonsense-mediated decay (NMD) or the exosome pathway. Therefore, IR coupled with RNA surveillance can regulate gene expression and affect various biological processes [3,4]. IR transcripts can be detained in the nucleus and wait for splicing signals, such as in the T cell activation process responding to external stimulus [5]. IR can also produce novel proteins with different functions [6,7,8] or subcellular locations [9], and play essential roles in many key biological conditions [3,10,11]. IR is under sophisticated regulation that can be affected by sequence variations, splicing factors, epigenetic and transcriptional regulatory mechanisms, etc. [12]

Tumors generally have 30% more aberrant alternative splicing events than normal tissues [13] and can give rise to many tumor-specific transcripts associated with oncogenic functions and drug resistance [14,15,16,17]. There are numerous relevant studies on exon skipping in cancer, while only a handful of them have identified features and functions of IR. Dvinge and Bradley reported that compared to adjacent normal tissues, more introns were retained in tumor tissues for 15 cancer types, and many of them could be detected in the cytoplasm [18]. In addition, there has been evidence that IR can promote cancer development. Somatic mutations in tumors can trigger IR and these mutations are enriched in tumor suppressor genes (TSGs) [19]. Inactivation of histone H3K36 methyltransferase *SETD2* reduced *DVL2* IR, and subsequently activated Wnt signaling to promote colon cancer predisposition [20]. Similarly, *ZRSR2* loss increased retention of a minor intron of *LZTR1*, which resulted in enhanced RAS signaling, potentially driving leukemia [21]. Of note, IR may produce epitopes presented on MHC I, making it a potential source of tumor-specific antigens (neoantigens) [22]. However, these studies have mainly focused on individual or patient-specific IR events, a systematic survey of recurrent IR alterations at a pan-cancer level, especially from the prognostic perspective, is still lacking.

In the present study, we profiled the IR landscape of 33 cancer types in the Cancer Genome Atlas (TCGA), identified IR events that were differentially regulated between normal and tumor samples, and discovered IR events associated with survival. Some informative introns not only showed potential in accurate diagnosis and prognosis with machine learning methods, but also were involved in cancer pathology, as validated in our experiments (Figure 1). Many of these introns were recurrently retained in multiple patients across cancers, serving as a resource of potential diagnostic and prognostic biomarkers, even as promising targets for new therapies.

## 2. Materials and Methods

### 2.1. RNA-seq BAM Download and IR Quantification

We downloaded RNA-seq BAM files using the GDC data transfer tool, including 33 cancer types and over 10,000 tumor and adjacent normal samples in the TCGA project. 

Stringtie (version 2.1.3) [23] was used to quantify gene expression. IRFinder (version 1.3.1) [2] was used to identify and quantify IR. Before applying IRFinder, samtools (version 1.9) was used to sort BAM files by read pairs. The genome version was hg38 and the gene annotation version was gencode v35.

We performed quality control on both intron and sample levels. Filtering out introns that overlap with known exons (flagged by IRFinder as “known-exon”) resulted in 243,151 introns covering 21,520 genes genome-wide. Before quantifying IR; at least 4 junction reads (split reads) spanning flanking exons were required to make sure that the isoform was expressed. IRratio was used to measure the IR level, and it was calculated by dividing the median read depth of an intron by itself plus the number of reads spanning flanking exons. If the coverage of an intron was less than 20%, it was likely to be completely spliced and its IRratio was assigned to 0. If the coverage of an intron was above 70% and the median read depth was above 3, the intron was likely to be retained and its IRratio was kept, which was greater than 0. Otherwise, the retention state could not be accurately decided for an intron and its IRratio was set as missing. IRFinder automatically decided if an RNA-seq sample was suitable for IR detection based on the ratio of the number of reads that map to intergenic regions to the number of reads that map to coding regions. When the ratio exceeded 10%, it gave a warning message, and such samples were not used for further analysis. Over 90% of TCGA RNA-seq samples passed this QC. Of note, acute myeloid leukemia (LAML) was under careful examination because the IR was prevalent across all LAML samples (Appendix A). The majority of TCGA RNA-seq was unstranded, so the read orientation could not be determined, which may lead to false positive IR detection [2,24]. Nevertheless, we made use of a small number of strand-specific RNA-seq samples in TCGA which passed sample quality control (*n* = 34) to generate a “whitelist” of retained introns, meaning they were more likely to be genuine and reliable. Specifically, we obtained a maximum IRratio of each intron in these samples, and introns with a maximum IRratio above 0.08 comprised the “whitelist” (*n* = 47,026). That is, these introns were retained in at least one of the stranded RNA-seq sample, which made them more reliable. Thus, differentially retained introns and survival-associated introns were restricted to only “whitelist” introns for higher confidence.

### 2.2. Differential IR and Differential Gene Expression

Because IR levels did not follow a normal distribution across individuals, we used paired Wilcoxon rank-sum test to detect differential IR events (DIRs) in paired tumor and normal samples (*n* > 15) for 14 cancer types. A differentially retained intron should have a *p*-value less than 0.05 and the difference of median IRratios between tumor and normal samples should be larger than 0.1.

In terms of differentially expressed genes (DEG) detection, DESeq2 [25] was used, and we selected DEGs with an adjusted *p*-value < 0.05 and |log2FC| > 1.

### 2.3. Dimensionality Reduction and Visualization

If the quality control mentioned above was not passed, the IRratio was set to a missing value. We generated IRratio matrices, kept IR events with missing rates less than 30%, and imputed missing values with a mean value from the remaining samples. Principle component analysis was then performed. We extracted the first 100 principal components and further analyzed them with package Rtsne (version 0.16) [26] to draw t-SNE plots, with perplexity set to 30.

### 2.4. Functional Enrichment

We used the package clusterprofiler [27] to enrich gene ontology (GO) biological process terms and KEGG pathways for target gene sets. GO terms related to cancer hallmarks were retrieved from two previous studies [28,29].

### 2.5. Sequence Features Analysis

We classified introns into four types to compare their sequence features. Constitutive introns (*n* = 48,344) were introns with median IRratio of 0 in tumor and normal samples across all cancers; unregulated IR were introns retained in tumor or normal samples in any cancer but showed no significant difference between tumor and normal samples (*n* = 10,887); down DIRs were introns with reduced retention level in tumor samples in any cancer (*n* = 401); up DIRs were introns with increased retention level in tumor samples in any cancer (*n* = 669).

Conservation analysis: hg38 version of phastCons30way.bw was downloaded from UCSC, and bwtool [30] was used to calculate a mean phastCons [31] score around boundaries of the above 4 types of introns separately.

Splice scores of 5′ and 3′ sites were calculated using the maximum entropy modeling method [32].

Intron GC content, length and relative gene position were all analyzed based on hg38 and gencode v35. To analyze the distribution of introns in genes, the relative gene position of an intron was represented by the percentile ranking of an intron among all the introns along a gene (5′ to 3′ orientation, and genes with fewer than 3 introns were excluded).

We adjusted a nonsense-mediated decay (NMD) prediction rule from a previous study [33]. Specifically, when an intron was retained in a protein coding gene, it was very possible to introduce a premature stop codon (PTC) and elicits NMD unless under following conditions. If the intron resides in a 5′ or 3′ untranslated region (UTR), or the intron was close to start codon (<200 nt), or it was the last intron, NMD would not be elicited. If no PTC was produced, or the PTC was located within 55 nt upstream of the last exon–exon junction, NMD would not be elicited either. Otherwise, the IR transcript was prone to be targeted by NMD.

### 2.6. Random Forests Model

We merged DIRs from different cancer types, and filtered out the ones with pan-cancer missing rates over 30% and the ones that were inconsistently up or down-regulated in different cancers. This resulted in 273 DIRs to be used in Random Forests models for pan-cancer modeling. R package randomForest (version 4.6-14) [34] was applied to train the model, with mtree = 500, mtry = 3 and proximity = TRUE. A hundred times four-fold cross validation was used to calculate a pooled area under the curve (AUC) for the training set (paired tumor and normal samples from 14 cancers), and the receiver operating curve (ROC) in a randomly selected one run was drawn using a R package pROC (version 1.18.4) [35]. We used the Rfcv function to predict model performance with a sequentially reduced number of DIRs (ranked by importance), with cv.fold = 5 and step = 0.9.

### 2.7. Survival Analysis

For introns that had valid IRratios in at least 50% of patients and at least 5% of valid IRratios were above 0.1, we restricted our analysis to patients with IRratios over 0 and classified them into IR-high or low groups based on median IRratio. Then, we performed Univariate Cox regression and selected the intron related to overall survival (OS) or disease-free survival (DFS), and an unadjusted *p*-value less than 0.05 was considered significant. Similarly, gene expression associated with survival was identified by selecting genes that with a median expression level over 1 TPM, dividing patients into high and low expression groups based on the median cutoff, and performing univariate Cox regression.

### 2.8. LASSO Regression to Build a Prognostic Model

To build an IR-based prognostic model for each cancer type, we selected introns with missing rates less than 20% (missing values were later filled with the mean) and performed LASSO regression with R package glmnet (version 2.0-8) [36,37]. The candidate introns and the corresponding coefficients were derived with the λ parameter associated with the minimum mean error or with one standard error. The intron retention risk (IRR) score was calculated as the sum of IRratios multiplied by corresponding coefficients of candidate introns. Patients were then divided into IRR low and high groups based on the median value.

### 2.9. Cell Culture and Lentiviral Transfection

HEK293 (human embryonic kidney 293 cells), H1299 cells, A549 cells were purchased from National Collection of Authenticated Cell Cultures and cultured in Dulbecco’s Modified Eagle’s Medium (DMEM) (Invitrogen, 11960044) supplemented with 10% FBS (Gibco), streptomycin (100 μg/mL), and penicillin (100 U/mL) at 37 °C/5% CO_2_. 

We applied lentivirus transfection-mediated gene-silencing strategy to stably knockdown target gene *STN1*. HEK293T cells were transfected as described [38]. Specifically, shRNAs (shRNA sequences were listed in Appendix A) were obtained from Sigma-Aldrich, then annealed and cloned into pLKO.1 vector. HEK293 cells grown in 6-well plates were transfected with 1 μg constructed vectors or empty control vectors pLKO.1 with VSVG and gag/pol encoding plasmids by using Lipofectamine 2000 (Invitrogen, Waltham, MA, USA). Then, the virus supernatant was harvested in 24 h to infect A549 cells in 6-well plates, that were seeded one day ahead. After incubation for one additional day, A549 cells were screened by 2.5 μg/mL puromycin for 24 h. The surviving cells were cultured for two additional days and then harvested for following RNA extraction and cell proliferation assays.

### 2.10. RNA Preparation, RT-PCR and qRT-PCR

Total RNA was extracted from cells with TRIzol (Invitrogen) according to the manufacturer’s instructions (Invitrogen, Waltham, MA, USA) and reversely transcribed into cDNA with oligo-(dT) primer (Appendix A) using FastKing RT Kit (Tiangen, Beijing, China, K116).

For RT-PCR, 100 ng cDNA was used for each PCR, and PCR products were detected by agarose gel electrophoresis. Gene expression at the RNA level was quantified by qRT-PCR using a 2× SYBR mix (Vazyme, Nanjing, China). GAPDH served as an internal control. Then, the reaction was run on the Bio-Rad CFX Manager. The IR and spliced transcripts of *STN1* were quantified by qRT-PCR using primers specifically targeting the retained intron and exon junction, respectively. All primer sequences are listed in Appendix A.

### 2.11. RNA Stability Assay, and Isolation of Nuclear and Cytoplasmic Fractions

A549 cells were treated with 10 µg/mL actinomycin-D (Act D; Sigma-Aldrich, Inc., St. Louis, MO, USA, A4262) for 0, 2, 4, and 6 h, respectively. Total RNA was extracted, and RNA levels were quantified by qRT-PCR as described above.

For nuclear and cytoplasmic fractionation, A549 cells were cultured in a T25 flask until reaching 90% confluence. Then, the cells were trypsinized, washed twice with cold PBS, and then fractioned by Nuclear/Cytosol Fractionation Kit (Phygene, Fujian, China, PH1466). Following the RNA extraction, qRT-PCR were carried out as described above.

### 2.12. Psi-CHECK2 Constructs and Dual Luciferase Assay

Luciferase reporter gene expression vectors were obtained from Promega and were prepared following to the manufacturer’s protocol. Briefly, the 590 bp intron-retained 5′ UTR and 146 spliced 5′ UTR was PCR-amplified using *STN1*_5′ UTR_Forward (Nhe I) 5′-TACGACTCACTATAGgctagcggggtcgtcgccgccag -3′ and *STN1*_5′ UTR_Reverse (Nhe I) 5’-TTGGAAGCCATGGTGgctagccaggctgcatcaagaggca-3′. PCR fragments were cloned into the NheI-restricted site of the psi-CHECK2 vector. The *STN1* 5′ UTR fragment was inserted directly upstream of the Renilla gene. The psi-CHECK2 construct containing the mutated 5′ UTR intron-retained fragment was synthesized by Tsingke Company, where the three start codons within the intron (181ATG, 283ATG, and 392ATG) were all mutated to AGC. DH5α cells were transformed with the three distinct constructs and cultured on ampicillin-containing media.

One day before transfection, 1 × 10^5^ HEK293T cells were seeded into each well of a 24-well culture plate in 500 μL DMEM supplemented with 10% FBS. Cells at 70% confluency were transfected with three psi-CHECK2 constructs with Lipofectamine 2000 reagent. Twenty-four hours after transfection, the growth media was removed and cells were washed gently with PBS. Passive lysis buffer (Promega, Madison, WI, USA) (100 uL/well) was added and gently shaken for 15 min at room temperature, then the cell lysates were harvested for dual luciferase assay. The activities of firefly and Renilla luciferase were measured using the Dual-Luciferase^®^ Reporter 1000 Assay System (Promega, Madison, WI, USA) as per the manufacturer’s protocol. A total of 25 μL of cell lysate was transferred into a white opaque 96-well plate. The luminescence obtained for both the mutated and wild-type constructs was normalized with the internal control firefly luciferase signal. Each experiment was performed in triplicate, and three independent experiments were performed. Quantitation of the reporter gene assay was calculated as mean ± SEM. A student’s *t*-test was used to determine significant differences between each mutated construct compared with the wild-type construct.

### 2.13. Colony Formation Assay, Transwell Migration Assay and Cell Proliferation Assay

For the colony formation assay, cells were cultured in the six-well plate at a density of 800 cells/well. Cells were cultured under normal culture conditions for 15 days. For fixation of the cell, after the medium supernatant was removed, the cells were treated with 4% paraformaldehyde and stained with 1% crystal violet (Sigma-Aldrich, St. Louis, MI, USA) for 15 min. Then, the plates were washed with phosphate-buffered saline (PBS) and photographed.

For the transwell migration assay, a 24-well transwell chamber (Corning, Corning, NY, USA) was used. Cells were suspended in non-serum DMEM and then seeded in the top chamber of the transwell with a density of 1 × 10^4^ per chamber, and 300 μL fresh complete DMEM (10% FBS) was added to the bottom chamber. After incubating for 48 h, using PBS to wash the cells in the top chamber twice, and then fix the cells with 4% paraformaldehyde for 15 min, and then stained with 1% crystal violet (Sigma-Aldrich) for 30 min. After washing and wiping off the cells on the inner side of the top chamber, the migratory cells adhering to the bottom surface of the membrane were observed, photographed and then counted by ImageJ software (Version 1.54).

Cell proliferation assay was performed on cultured cells at four time points (24, 48, 72, and 96 h, respectively). A total of 100 μL of cells were seeded in a 96-well plate with 1000 cells/well minimum and four replicates for each time point. Cell Counting Kit-8 (CCK-8) reagent (Dojindo Laboratories, Kumamoto, Japan) was added according to the manufacturer’s protocol. Then, cells were incubated at 37 °C for 2 h and the absorbance at 450 nm were measured with a microplate reader (TECAN).

## 3. Results

### 3.1. Landscape of Intron Retention in 33 Cancer Types

RNA-seq data (in BAM format) of the primary tumor and adjacent normal tissues from 33 TCGA cancer types, were downloaded and processed with IRFinder [2]. A total of 10,189 samples passed quality control and were used in this study (Figure 2A, Appendix A). IRratio was used to measure the retention level of each intron. The median IRratios of each intron among the tumor samples were used to represent the IR level in each cancer type, and the correlation coefficients between different cancers were generally higher than 0.6. While acute myeloid leukemia (LAML), esophageal carcinoma (ESCA), and stomach adenocarcinoma (STAD) had lower similarity IR patterns with the other cancer types (Figure 2B). LAML was the only hematological tumor (liquid biopsy) among all test tumors. Dvinge and Bradley have observed the strongest IR increase in LAML among all cancers [18], and the average number of retained introns detected in LAML was also the highest in our analysis (Appendix A). As for ESCA and STAD, they had the highest average sequencing depth, and both had a read length of 75 bp, compared to 48 bp for most cancer types (Appendix A). This may be one possible cause for why they had lower IR similarities with other cancers.

Next, we compared the average number of retained introns (IRratio > 0.1) between the tumor samples and normal samples (Figure 2C, Appendix A). In total, 13 cancer types exhibited significantly increased numbers of retained introns compared to normal samples, while breast invasive carcinoma (BRCA) showed the opposite trend. Our results validate the findings of Dvinge and Bradley, reinforcing that inefficient intron removal is a common phenomenon in many cancers, with the exception of breast cancer.

### 3.2. Differentially Retained Introns between Tumor and Normal Tissues

To accurately compare IR between normal and tumor samples, we analyzed 14 cancer types each, with at least 15 paired normal samples and tumor samples (Appendix A). Principal component analysis (PCA) showed varying degrees of differences in global IR patterns for normal and tumor samples across cancer types (Appendix A). We used a paired Wilcoxon rank-sum test to detect introns significantly upregulated or downregulated in tumor tissues compared to their matched adjacent normal tissues for each cancer type (*p* < 0.05) and required differences in median values between normal samples and tumor samples greater than 10%.

A total of 988 differential introns were detected across 14 cancers, most of which were retained in both normal and tumor samples but with altered intron retention levels (Appendix A), while a handful of them were mainly retained in tumor samples and spliced in normal samples (cancer-associated IR), or the other way around (i.e., normal-associated IR) (Figure 3A). Colon adenocarcinoma (COAD) and BRCA were detected with the most differential IR events (DIRs), with most DIRs upregulated in COAD and downregulated in BRCA. DIRs were often specific to only one cancer type (Appendix A), and DIR genes usually had only one intron that was differentially retained (Appendix A). The IR alteration between tumor and normal samples demonstrated a weak negative correlation with mRNA expression change (R = −0.17, *p* = 2.7 × 10^−11^, Spearman correlation), and on average, less than 15% of DIR genes were differentially expressed, according to the changes in IR (Figure 3B). Their mild correlation reflects the known function of IR to fine-tune the gene expression through RNA degradation [3,5,10,11,39,40,41].

Mutation-induced IR has been widely reported to inactivate tumor suppressor genes (TSGs) [19]. However, such mutations were found in only a minority of patients. In contrast, DIRs affected multiple patients. Although not enriched in COSMIC TSGs or oncogenes [42], DIRs were found in some cancer-related genes. For example, retention of intron 15 of *LZTR1*, a TSG that negatively regulates RAS signaling through ubiquitination [43,44,45], was upregulated in COAD, STAD, and uterine corpus endometrial carcinoma (UCEC) tumor samples (Appendix A). Although the expression of *LZTR1* did not significantly change between tumor samples and normal samples, elevated IR still implies a decreased level of spliced and functional products. A minor intron retention of *LZTR1* is associated with tumorigenesis in leukemias [21]. Another affected TSG was *ERCC4* (Appendix A), which functions in nucleotide excision repair [46,47]. Some oncogenes were also affected by DIRs. Tumor samples of BRCA exhibited lower retention of intron 3 in *CSF3R* (Appendix A), which is a highly mutated oncogene in chronic myeloid leukemia [48,49]. DIRs affect various biological pathways in different cancers, such as DNA damage and cell cycle checkpoint in lung squamous cell carcinoma (LUSC) (Appendix A). These results indicate that DIRs detected in multiple patients were found in genes with various biological functions, including those relevant to carcinogenesis, though their functional consequences and detailed mechanisms require further studies.

### 3.3. Differentially Retained Introns Were Shorter in Length

Many sequence features that are known to predispose introns to retention were also discovered in our retained introns, including higher GC content, shorter intron lengths, biased distribution at the 3′ end of the gene, and lower likelihood of inducing NMD (Figure 3C,D and Appendix A). Notably, introns differentially retained in cancers were shorter than unregulated ones (the median intron lengths of the downregulated, upregulated, and unregulated IR were 456, 452, and 934 bp, respectively) (Figure 3C). We speculate that shorter introns may have fewer RBP-binding sites, and thus may be more sensitive to RBP expression changes during oncogenesis. However, no significant difference in splice signal strength between different groups of introns (Figure 3D). This is somewhat puzzling, as many previous studies have found retained introns to typically have weaker splice sites [3,18,50]. However, Zhang and colleagues have also reported that introns differentially retained between prostate cancer and normal tissues have splice signals comparable with control introns [51]. Retained and constitutive introns demonstrated almost identical conservation within the intronic boundaries sequence, again supporting their comparable splice signals strength. However, constitutive introns had the most conserved flanking exonic boundaries, followed by down and up DIRs, indicating their higher selective pressure than unregulated retained introns (Appendix A).

We also compared the sequence features of up-regulated and down-regulated introns in each cancer type. The results suggested that some features may underlie the regulation of DIR in specific cancers (Appendix A). For example, compared to down-DIRs, COAD and liver hepatocellular carcinoma (LIHC) both exhibited weaker 3′ splice signals in up-DIRs compared to down-DIRs. This may explain the increased retention levels observed in these introns. In addition, higher GC content, known to predispose introns to retention, was observed in up-DIRs in several cancers, including lung adenocarcinoma (LUAD), LUSC and UCEC.

On the other hand, potential DIRs induced by somatic variants can be identified by many tools [52,53,54]. Mutations near splice site in general are more likely to affect splicing and we therefore examined whether somatic mutations were present within 20 bp distance of the splice sites of DIRs in the studied patient samples. The results showed that most DIRs did not have mutations near the splice sites, only 20 DIRs had mutations detected in a single patient. However, DIRs were obtained by intergroup comparisons between cancer and normal tissues, that is, the trends in DIRs were common across multiple patients, and the presence of somatic mutations in individual patients does not explain the regulation of DIRs. In summary, somatic mutation does not appear to be a major cause of DIR.

### 3.4. Differential IR Events Showed Diagnostic Potential

We used t-SNE [26] to visualize IR patterns in over 10,000 samples from 33 TCGA cancer types. Genome-wide introns (*n* = 37,845) had limited tissue origin specificity and tumor vs. normal specificity (Figure 3E,F), whereas DIRs (*n* = 375) exhibited greater specificity (Figure 3G,H). Similar results were observed when restricting the analysis to just the 14 cancer types with detected DIRs (Appendix A). These results suggest DIRs may distinguish tumor and normal samples. We therefore applied Random Forests [34] to assess the potential of IR as cancer biomarkers. 

After removing DIRs that were inconsistently upregulated and downregulated in different cancer types and DIRs that had a missing rate above 30% in all samples, 273 DIRs were left for model training. All TCGA samples were divided into three groups: (1) a training set including 1210 paired tumor and normal samples from 14 cancers, which were the same samples used for DIRs detection; (2) the first validation set including 5738 unpaired tumor and normal samples from the same 14 cancers; (3) the second validation set including 3190 samples from the rest 19 cancer types (Figure 4A). A hundred times four-fold cross validation with the training set yielded a pooled area under curve (AUC) of 0.958, and the result of one randomly selected run was shown in Figure 4B. When using the entire training set to train the model, the AUCs for the first and second validation sets were 0.936 and 0.838, respectively (Figure 4C,D). In addition, comparable performance was achieved using just the top 30 DIRs (Figure 4E,F). Two representative DIRs are displayed here, which were downregulated (*ZDHHC8*) and upregulated (*ZNF800*) in five cancer types, respectively (Figure 4G,H). We also test the utility of DIRs as biomarkers for each cancer type, and the AUC of Random Forests models was all above 0.9 (Appendix A). Taken together, DIRs demonstrate powerful diagnostic potential in a pan-cancer level and for a specific cancer type.

### 3.5. Identify Prognostic IR Events across Cancers

We expanded our analysis to all primary tumor samples from 33 cancers in TCGA to explore the prognosis potential of introns with varying degrees of retention in patients. Specifically, we investigated introns that had a valid IRratio for over 50% of patients within a cancer type. The IRratio must have been > 0.1 in at least 5% of these patients. Then, univariate Cox regression was applied to patients with IRratio > 0. The results revealed over 100 IR events associated with overall or disease-free survival (*p* < 0.05) in 29 cancer types. Among these, LAML had the highest number of prognostic IR events (Figure 5A). Notably, the prognostic IR events for some cancers, such as LAML and prostate adenocarcinoma (PRAD), were predominantly associated with either favorable or adverse outcomes. A total of 11,522 prognostic IR events were detected among 33 cancers, of which 67.4% were specific to one cancer type, and 32.6% were shared by different cancers (Figure 5B, Appendix A).

Noteworthy, the number of prognostic IR events was not proportional to the number of genes whose expression was associated with survival (Appendix A). Moreover, only ~19% of prognostic IR genes had expression levels also associated with prognosis (Appendix A). The discordance suggests that IR may have unique prognostic features independent of gene expression. On average, 42.8% of prognostic introns were negatively correlated with RNA expression across all cancer types, with 6.5% strong negative (r ≤ −0.5) association (Appendix A).

### 3.6. Prognostic Introns Affect Genes Involved in Tumorigenesis

Gene Ontology (GO) terms enriched for prognostic IR genes from 33 cancer types include chromatin organization, histone modification, cell cycle arrest, and GTPase activity regulation (Figure 5C). From the perspective of individual cancer types, 25 cancer types showed prognostic IR genes enriched for GO terms associated with cancer hallmarks (Figure 5D; Appendix A). Sustained angiogenesis, tissue invasion and metastasis, and genome instability and mutation were the three most frequently altered hallmarks. Prognostic IR genes were also enriched in cancer-related KEGG pathways, including MAPK signaling pathway and the PD-1 checkpoint pathway (Appendix A). Compared with non-prognostic IR genes, prognostic IR genes were significantly enriched for COSMIC cancer genes in lymphoid neoplasm diffuse large B-cell lymphoma (DLBC), kidney renal papillary cell carcinoma (KIRP) and LAML (Appendix A). The increased last intron retention of *MYC*, a famous oncogene, was associated with a favorable prognosis in BLCA and negatively correlated with expression (Appendix A). One interesting example we found is the gene *STN1*, encoding a subunit of CST (CTC1-STN1-TEN1) that plays a key role in telomere maintenance. Intriguingly, the retention level in the 5′UTR of *STN1* was decreased in tumor samples compared with normal samples, and lower retention level is significantly associated with poorer survival outcomes in LUAD (Figure 6A,B and Appendix A). However, its mRNA level was not correlated to patient survival, and the level of IR and mRNA were not correlated (Appendix A). This implies that this IR event may impact patient survival independent of *STN1* mRNA expression. We further examined this 5′ UTR intron in the A549 lung cancer cell line and found that the retention level of this intron was high in the A549 cell line, with a ratio of spliced transcripts to IR transcripts of approximately 2:1. (Figure 6C,D). Therefore, A549 was used as our experimental model, as high inherent IR levels facilitate experimental manipulation and IR detection.

The RNA stability assay in A549 demonstrated that *STN1* IR transcripts had a similar degradation rate as the spliced transcripts, consistent with the lack of correlation between IR and mRNA levels observed in TCGA LUAD samples (Figure 6E and Appendix A). In addition, nuclear and cytoplasmic fractionation revealed that a greater proportion of the IR transcripts were detained in the nucleus compared to spliced transcripts (Figure 6F). On the other hand, retention of this intron extends the length of the 5′ UTR and produces three potential upstream open reading frames (uORFs) (Appendix A). uORFs are known to downregulate protein expression by decreasing translation efficiency [55]. We next investigated the effect of this uORF-containing 5′ UTR on translation using a dual luciferase assay (Appendix A). We used psi-CHECK2 plasmids with three different *STN1* 5′ UTR sequences inserted upstream of the Renilla coding region: the spliced 5′ UTR (shorter), the intron-retained 5′ UTR (longer), and the intron-retained 5′ UTR with start codon ATGs mutated to AGCs (longer and mutated) (Appendix A). The results showed that compared to the spliced *STN1* 5′ UTR, the 5′ UTR IR significantly inhibited translation, while mutating start codons of all three uORFs nearly restored translation efficiency (Figure 6G). These results demonstrated that the retention of the first intron of *STN1* reduced its protein production through the production of uORFs, along with the nucleus detention of IR transcripts.

Because the intron retention of the *STN1* 5′ UTR may downregulate STN1 protein level, knocking down *STN1* should mimic the functional impact of increased IR levels, which is seen in adjacent normal tissues and in tumor tissues from patients with better prognosis. Since *STN1* is known to be essential for telomere replication and cell proliferation [56], knocking down *STN1* in A549 did significantly reduce levels of known proliferation markers including CDK1 and MKI67 (Figure 6H). Furthermore, clonogenicity, cell motility, and proliferation were impaired in *STN1*-knockdown A549 cells (Figure 6I,J).

Taken together, the identified IR event in the *STN1* 5’ UTR directly regulates STN1 protein production through uORF-mediated translation inhibition and nucleus detention (Figure 6K), which in turn impacts tumor cell proliferation, clonogenicity and motility, ultimately influencing patient survival.

### 3.7. IR Enables Accurate Risk Stratification in Multiple Cancers

The current standard of care assessment for many cancers relies on clinical biomarkers and the detection of specific genetic mutations or chromosomal structural variation. For example, in LAML, cytogenetic screening and target gene sequencing-based stratification are effective for half of LAML patients. The rest of the patients are difficult to assign to a defined risk group since they are cytogenetically normal and do not carry mutations in known risk genes [57]. Therefore, there are ongoing efforts to improve the risk stratification for LAML and other cancers by incorporating additional molecular features into prognostic models.

Because LAML has the highest number of prognostic introns, we first tried to build an IR prognostic model with the least absolute shrinkage and selection operator (LASSO) regression [36] for LAML. The resultant model was made up of four IR signatures and was designated as an IR risk (IRR) (Figure 7A, Appendix A). Based on the IRR score, we can divide the LAML patients from TCGA into two risk groups with significantly diverged overall survival outcomes (HR = 5.24, *p* = 1.36 × 10^−10^, Figure 7B). We used univariate Cox regression to compare the contribution of IRR and other reported clinical and molecular predictors [57,58], and IRR turned out to be the most powerful predictor (Figure 7C, Appendix A). Although age over 60 at diagnosis is a known risk for LAML, IRR still enabled effective stratification in patients under or over 60 years old (Figure 7D).

Similarly, we were able to construct a prognostic model based on 4~30 IR events with LASSO regression for 32 cancer types, except for testicular germ cell tumors (TGCT), achieving good risk stratification (Figure 8). This demonstrates the potential of IR as powerful prognostic markers for many cancers.

## 4. Discussion

Aberrant AS is a hallmark of cancer, and tumor tissues have about 30% more AS events than normal tissues [13]. By producing tumor-specific proteins or by altering the production of normal proteins, aberrant AS can lead to the activation of proto-oncogenes or the inactivation of TSGs, ultimately affecting cell growth and differentiation, angiogenesis, tissue invasion, and metastasis [59]. Therefore, the study of aberrant splicing not only helps to understand the mechanisms of cancer initiation and development but also has potential clinical implications [60]. There is an intriguing imbalanced IR pattern in multiple cancers where tumors exhibit a significant increase in IR compared to normal samples, which is not observed in other AS types [18]. IR may contribute to cancer development by inactivating TSGs in cancer patients [19]. Moreover, IR can encode novel proteins and may be an important source of tumor-specific antigens [22]. In this study, we systematically quantified and profiled IRs in 33 cancers from TCGA and tentatively explored their clinical relevance. To our knowledge, this is a comprehensive analysis of IR regarding the largest number of cancer types.

We identified differential intron retention events (DIRs) by comparing paired tumor and normal tissues in multiple cancer types. We found that the splicing signals of differentially retained introns were almost as strong as constitutive introns, which was consistent with Zhang et al.’s finding [51]. One possible explanation is that DIRs were recurrent in tumor and/or normal samples, and they may have similar biological importance as constitutive introns. Interestingly, DIRs were shorter than both constitutive introns and unregulated retained introns. Zhang et al. have reported that shorter exons are more likely to be excluded in cancers and are possibly regulated by elevated transcription and dysregulation of some RBPs in cancer cells [61]. In addition, most genes are spliced co-transcriptionally [62], and increased RNA Pol II accumulation in retained introns has also been reported [3,63]. Whether shorter introns are more sensitive to transcription and other splicing alterations in cancer requires further investigation.

Compared with adjacent normal tissues, dozens to hundreds of introns per cancer showed up or down-regulation in tumor tissues, resulting in a total of 988 DIRs across 14 cancer types. Some of them were cancer-type specific (such as *CSF3R*), while some others (such as *LZTR1* and *ERCC4*) showed consistent alterations across multiple cancers. We further identified 30 DIRs that stratified tumor samples and normal samples, demonstrating their diagnostic potential. A recent study exerted intron splicing events generated by *SF3B1* mutations that are specific to tumor patients to design synthetic introns and achieve targeted clearance of tumor cells [64]. Similarly, DIRs in our study may also serve as promising candidates for therapeutic synthetic introns, since they were widespread in multiple patients and even in multiple cancers (e.g., IR of *ZDHHC8* in Figure 4G). On the other hand, because retained introns can also encode peptides located on cancer cell surfaces [22], commonly retained introns may be potential targets for “off-the-shelf” immunotherapy.

We discovered several introns with the potential to indicate survival outcomes. For example, we experimentally validated a functional prognostic intron in LUAD. Specifically, the 5′ UTR intron regulates the translation efficiency of *STN1*, which is essential for cancer cell proliferation through maintaining telomere replication and genome stability. These insightful results also indicate potential novel therapeutic strategies to combat LUAD.

There have been ongoing efforts in exploring new biomarkers for risk stratification in cancers, such as gene expression [58,65,66,67,68]. However, splicing has been reported to outperform gene expression analysis in predicting survival in multiple tumor types [69,70]. The potential of alternative splicing (AS) in prognosis has also been demonstrated in various cancers, including ovarian cancer [71], colorectal cancer [72], lung cancer [73], esophageal cancer [74], liver cancer [75] and adrenocortical carcinoma [76]. We explored the prognostic power of IR in 33 cancer types, and most of the cancers (*n* = 32) can be stratified with less than 30 introns. The performance of the IR-based prognostic model in TCGA LAML cohorts outperformed clinical and other molecular predictors. IR has been reported to indicate prostate cancer aggressiveness [51] and pancreatic cancer clinical outcomes [69,77]. Our results further suggest that IR can serve as an accurate and powerful biomarker in multiple cancers.

With advances in technology, many types of biomarkers in blood have been found to have diagnostic and prognostic capabilities, including circulating tumor cells (CTC), circulating tumor DNA (ctDNA), and cell-free RNA (cfRNA). Among them, cfRNA is a promising type of biomarker due to the large number of dynamic changes found between cancer tissues and normal tissues, as well as among cancer patients, including differential expression and post-transcriptional regulations (such as alternative splicing). cfRNA detection has the advantage of being non-invasive, can be detected by cost-effective RT-qPCR, and is therefore advantageous for a wide range of clinical applications. Promising applications for diagnosis and prognosis of cfRNA has been demonstrated in a variety of cancers, including liver cancer, myeloma, colorectal cancer, breast cancer, lung cancer, etc. [78,79,80,81] A number of differential and prognostic IR events with tissue specificity were found in our study, some of which may have the potential to predict tumor tissue origin. Through subsequent rigorous screening and validation in plasma from tumor and normal samples, IR biomarkers may be applicable in liquid biopsy.

DIRs and prognostic IR events were two types of informative IR characterized in this study. Differentially retained introns across many cancer types were heterogenous and influenced genes of various functional categories. Prognostic introns affected genes involved in cancer-related pathways, including DNA damage and cell cycle regulation, angiogenesis, cancer cell invasion and metastasis. In DLBC, KIRP and LAML, prognostic IR genes were significantly enriched for COSMIC cancer genes. IR has been recognized as a widespread mechanism of TSG inactivation [19], and we found that IR may also regulate the activity of oncogenes, such as *MYC* (Appendix A).

IR detection through poly(A)-enriched RNA-seq may be affected by unspliced pre-mRNA [82,83]. However, it is technically difficult to distinguish between the real IR signal and the pre-mRNA interference with regular RNA sequencing methods. To the best of our knowledge, most studies based on RNA sequencing have assumed that reads from polyA-enriched RNA-seq are from mature mRNA. Most IR studies followed such an assumption and have identified many IRs in various biological samples, some of which showed critical significances in many physiological and pathological situations [5,38,41,84]. This assumption is further supported by the correlation between IR level and expression level of its host gene in our study. If most IR signals came from pre-mRNA, then the IRratio should reflect the relative ratio of nascent RNA to mature RNA and an increase in this ratio should also indicate an increase in transcriptional activity. Then, a positive correlation between IRratio and expression level is expected under this hypothesis. However, throughout the 33 cancers in TCGA, the proportion of prognostic introns that were positively associated with host gene expression was much smaller than negatively associated or not associated ones (Appendix A). Since many IR transcripts are prone to degradation (e.g., through NMD pathway) and some do not affect RNA stability, IRratio is expected to negatively or not correlate with host gene expression, which is consistent with our observation. Therefore, the intron signals detected in the present study are more likely to come from retained intron rather than from pre-mRNA.

Of note, many IR transcripts have been reported to be located in both nucleus and cytoplasm, which is the case for our experimentally validated IR of *STN1* (Figure 6F), so even RNA-seq after nuclear/cytoplasm separation cannot separate IR transcripts from pre-mRNA.

lncRNAs with a polyA tail and overlapping with introns of other genes are also an interfering source of IR study. We found that only a very small fraction (5% differentially retained introns and 7% prognostic introns) of retained introns overlapped with annotated lncRNA exons. In addition, lncRNAs have much lower expression levels than mRNAs [85,86]. Taken together, we believe that the influence of lncRNA on IR should be very small. Using stranded RNA-seq can further decrease anti-sense RNA noise in IR studies.

One of the limitations of our study is that we still lack experimental assessments of the functions of a large number of informative IR events we identified, since IR transcripts have diverse fates [12]. The second limitation is that only a few patients in the TCGA dataset have matched normal tissues. More normal samples as backgrounds or negative controls will contribute to finding more informative IR events in future analyses.

## 5. Conclusions

Overall, we systematically characterized IR at a pan-cancer level, explored its clinical relevance, and provided experimental evidence of IR in cancer pathology. Our results revealed a rich resource for IR that had potential clinical applications, including cancer biomarkers and even therapeutic targets.

## Figures and Tables

**Figure 1 cancers-15-05689-f001:**
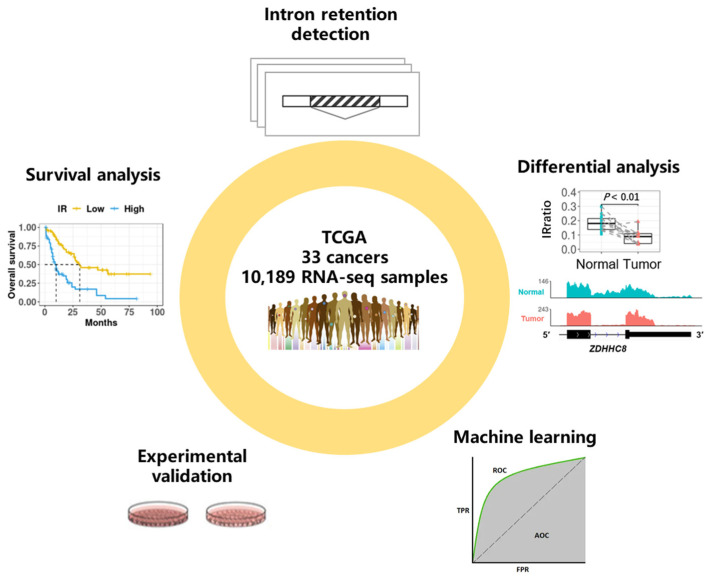
Schematic overview of data set and analysis. We downloaded over 10,000 RNA-seq samples originating from 33 cancer types from the TCGA data portal and detected genome-wide intron retention events for each sample. Differential analysis between adjacent normal tissues and cancerous tissues was conducted to find differential IR events, and survival analysis was conducted to find prognostic IR events. We demonstrated that some informative introns were potential diagnostic or prognostic biomarkers with machine learning methods. Importantly, our molecular and functional experiments validated that IR plays a role in cancer pathology.

**Figure 2 cancers-15-05689-f002:**
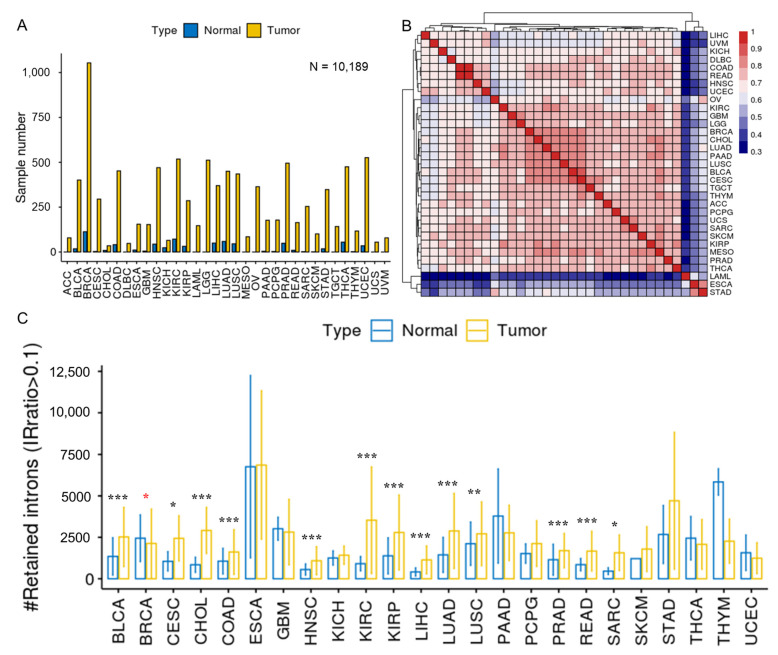
Overview of intron retention across 33 cancer types. (**A**) Sample statistics for a total of 10,189 samples from 33 cancer types. Only samples that passed IRFinder QC were analyzed. (**B**) Hierarchical clustering of median IRratio across cancer types based on pair-wise Spearman correlation. (**C**) The average number of retained introns (IRratio > 0.1) in tumor and normal samples for each cancer type (cancers without normal sample were excluded). Error bars indicate standard deviation. *, *p* < 0.05; **, *p* < 0.01; ***, *p* < 0.001; two-tailed student’s *t* test. Red asterisks indicate that more introns are retained in normal samples than in tumor samples and black asterisks indicate the opposite.

**Figure 3 cancers-15-05689-f003:**
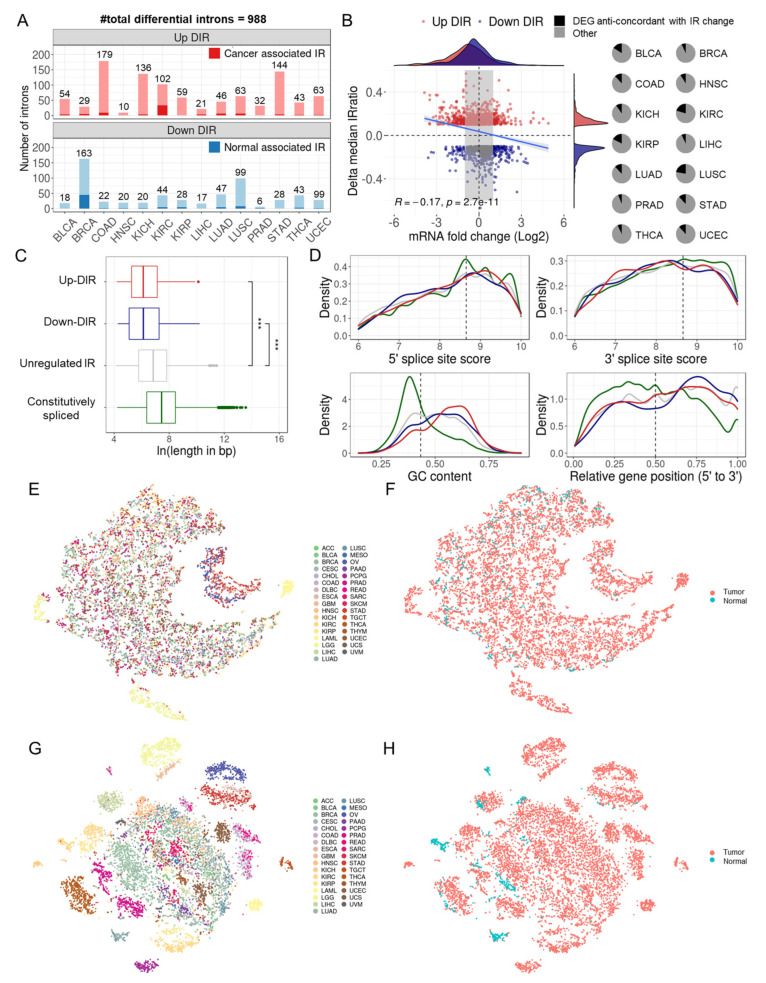
Differential IR events between tumor and adjacent normal samples. (**A**) Statistics of upregulated and downregulated differential IR events (DIRs) in 14 cancer types. Cancer associated IR referred to introns that tend to be spliced in normal samples (median IRratio = 0) and retained in tumor samples (median IRratio > 0.1). The opposite is normal associated IR. (**B**) The relationship between DIRs and gene expression changes. The left panel shows a comparison of median IRratio change and corresponding mRNA fold change between tumor and normal samples. Grey area masked genes whose expression fold change were between 1/2 and 2. The right panel shows the proportion of differentially expressed genes that were anticorrelated with IR change (i.e., expression increases and IR decreases, and vice versa). (**C**) Length distribution of four types of introns: upregulated, downregulated, unregulated retained introns, and constitutively spliced introns. ***, *p* < 0.001, two-tailed Mann-Whitney U test. Four colors indicate four types of introns. (**D**) Comparison of GC content, relative position in genes (the percentile ranking of an intron along a gene) and splice signals across four types of introns. The colors of the lines represent different introns, consistent with (**C**). Dashed lines indicate median values across all introns throughout the genome. (**E**,**F**) t-SNE plot of genome-wide introns (*n* = 37,845 after missing rate filter) colored by cancer types (**E**) or by tumor or normal sample (**F**). (**G**,**H**) t-SNE plot of DIRs (*n* = 375 after missing rate filter) colored by cancer types (**G**) or by tumor or normal sample (**H**).

**Figure 4 cancers-15-05689-f004:**
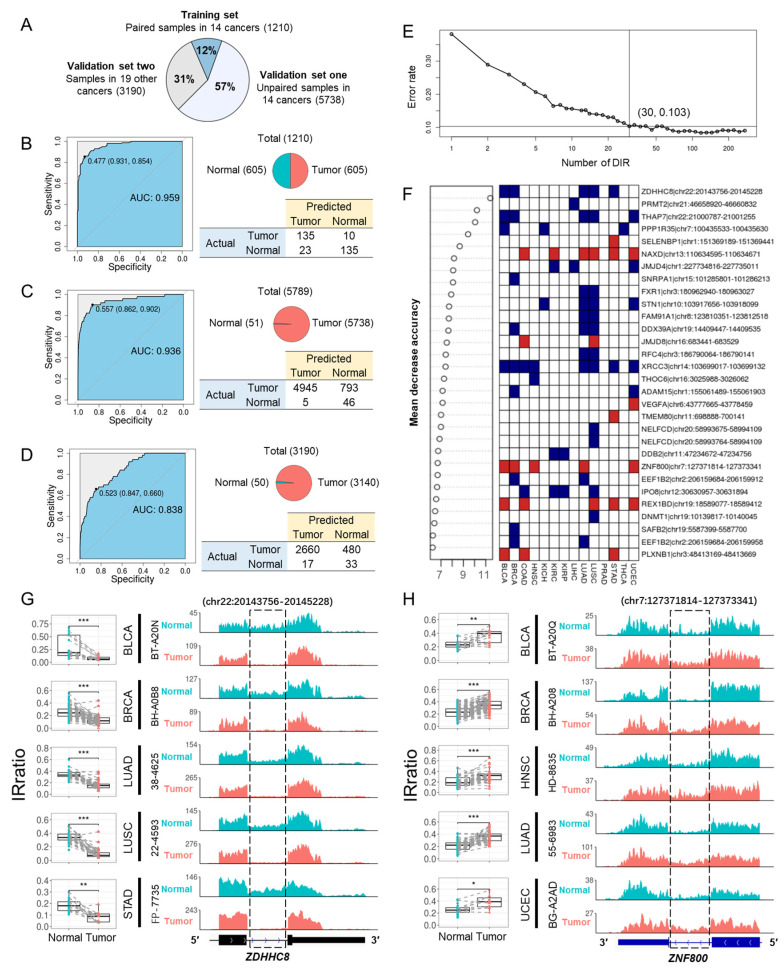
Differentially retained introns can distinguish tumors from normal samples. (**A**) TCGA samples were divided into three groups: paired tumor and normal samples from 14 cancer types that were previously used to detect DIRs (training set for Random Forests model, *n* = 1210), unpaired samples in the same 14 cancer types (validation set 1, *n* = 5789), and all samples from 19 other cancer types (validation set 2, *n* = 3190). (**B**) Random Forests model performance in training set in a four-fold cross validation run. The upper right panel shows the sample distribution. The left panel showed the ROC of the model and the bottom right panel shows the confusion matrix. (**C**,**D**) Random Forests model performance in validation set 1 (**C**) and set 2 (**D**) when using all 1210 paired samples from 14 cancer types to train the model. (**E**) Five-fold cross-validated prediction performance of Random Forest models in training set with a sequentially reduced number of DIR (ranked by importance). (**F**) Top 30 DIRs that contributed most to model accuracy and their differential retention in 14 cancer types. Blue and red denote down and up-regulated introns, respectively. (**G**,**H**) *ZDHHC8* (**G**) and *ZNF800* (**H**) final introns showed differential retention in paired tumor (red) and normal (blue) samples in five cancer types. The left panels were IRratio boxplots and the right panels were RNA-seq read coverage from example patients. *, *p* < 0.05; **, *p* < 0.01; ***, *p* < 0.001; paired Wilcoxon rank-sum test.

**Figure 5 cancers-15-05689-f005:**
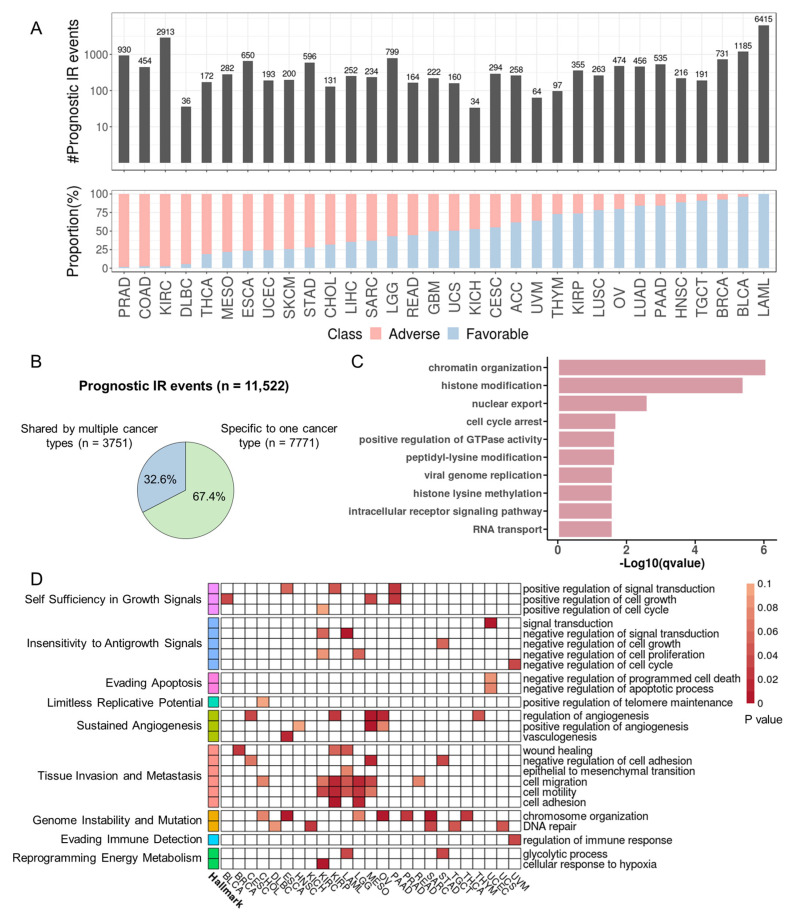
Statistics and functional enrichment of prognostic introns. (**A**) Statistics of prognostic introns. The upper panel shows the number of prognostic introns for each cancer type. The bottom panel shows the percentage of prognostic introns associated with longer (favorable prognosis, blue) or shorter (adverse prognosis, red) survival for each cancer type. (**B**) Proportions of cancer-specific and pan-cancer prognostic introns. (**C**) GO term enrichment analysis of genes with prognostic IR. (**D**) Enrichment of prognostic IR genes in each cancer for GO terms related to cancer hallmarks. GO terms (rows) were sorted according to cancer hallmarks on the left. Cell color indicates *p*-values from hypergeometric test; white cells indicate *p* > 0.1).

**Figure 6 cancers-15-05689-f006:**
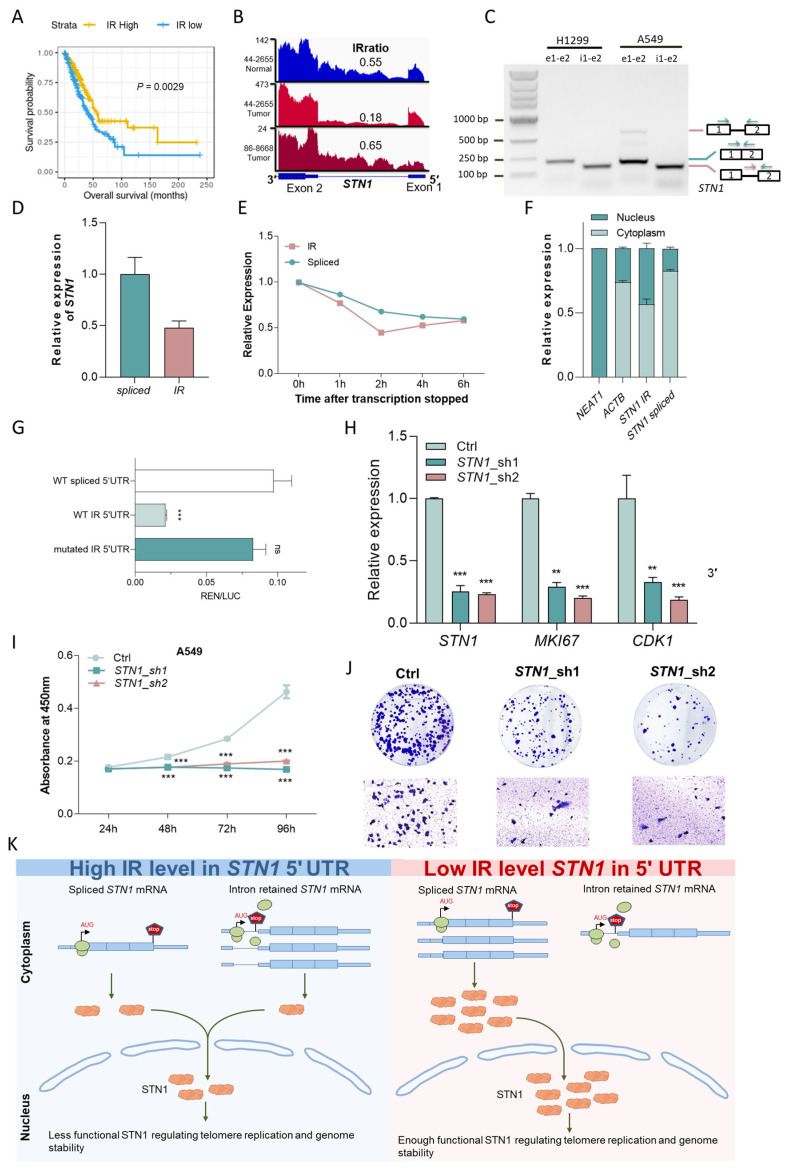
A 5′UTR IR event reduces the translation efficiency of *STN1* and may suppress tumor growth. (**A**) Increased IR in 5′ UTR of *STN1* was associated with longer survival in lung adenocarcinoma patients. (**B**) RNA-seq read coverage from a matched normal sample and tumor sample and another tumor sample. (**C**) RT-PCR using primers amplifying exons 1–2 (‘e1-e2′) as well as specific to IR isoform in H1299 and A549 cells. (**D**) Relative intron retention levels of *STN1* in A549 cells detected by qRT-PCR. (**E**) Relative expression of IR and spliced transcripts of *STN1* detected by qRT-PCR after inhibition of transcription in actinomycin D (10 µg/mL)-treated A549 cells. (**F**) qRT-PCR for *STN1* spliced and IR transcripts, following nuclear and cytoplasmic fractionation of A549 cell lysates. *NEAT1* and *ACTB* serve as nuclear and cytoplasmic markers, respectively. (**G**) The translation efficiency of three 5′ UTRs was detected by the relative Renilla/Firefly fluorescence intensity ratio. ***, *p* < 0.001; two-tailed student’s *t* test. (**H**) Expression levels of *STN1*, *CDK1*, *MKI67*, were detected by qPT-PCR upon *STN1* knockdown. **, *p* < 0.01; ***, *p* < 0.001; two-tailed student’s *t* test. (**I**) The cell proliferation rate of A549 cells evaluated by CCK-8 assay upon *STN1* knockdown. ***, *p* < 0.001; two-tailed student’s *t* test. (**J**) Effects of *STN1* knockdown on cell viability in the clonogenic assay (**top**) and cell migration assay (**bottom**). (**K**) Working model for *STN1* 5′ UTR intron retention in regulating cancer cell survival. Decreased *STN1* IR levels in LUAD patients leads to increased production of functional STN1 protein, which regulates telomere replication and genome stability, ultimately aids cancer cell survival and proliferation.

**Figure 7 cancers-15-05689-f007:**
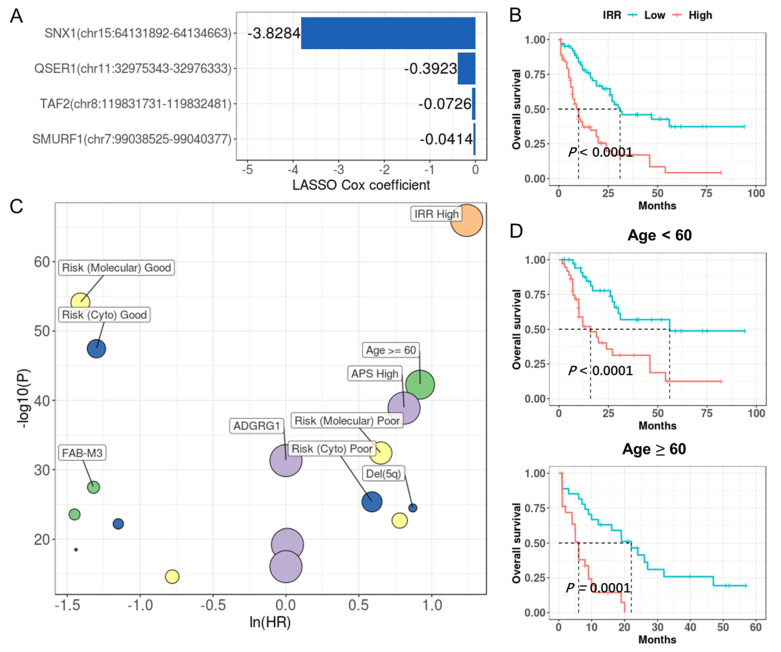
IR prognostic model stratifies patients in LAML. (**A**) IRR model coefficients. *Y*-axis indicates introns and corresponding genes that were used to build this model, and the *x*-axis indicates their LASSO Cox coefficients. (**B**) Kaplan–Meier curves of high and low risk groups divided based on median IRR. *p* values were calculated with a log rank test. (**C**) IR (orange), clinical (green), mutational (yellow), cytogenetic (blue), and expression (purple) features that were significantly associated with prognosis in univariate COX regression. *X*-axis and *y*-axis indicate hazard ratios and the corresponding *p* values, respectively (both were log-transformed). Point size reflected the percentage of patients affected by the investigated feature. (**D**) IRR enabled effective stratification in patients under or over 60 years old.

**Figure 8 cancers-15-05689-f008:**
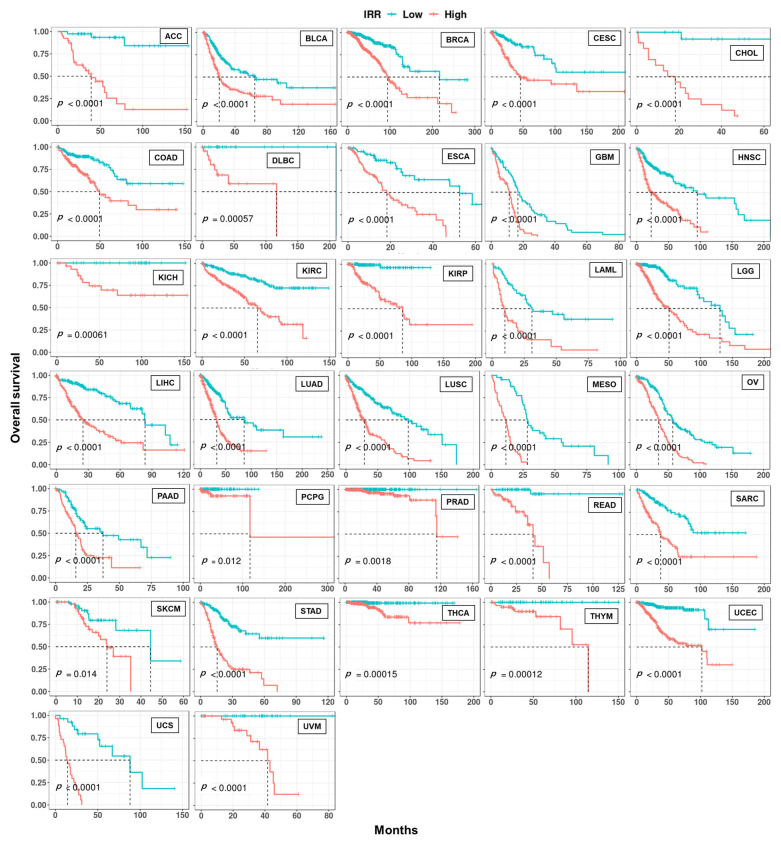
Kaplan–Meier curves of IR-based prognostic models in 32 cancer types. For each cancer, LASSO regression was performed to build an IR-based prognostic model which we called IRR. Patients in each cancer type were classified into high and low risk groups based on median IRR, and Kaplan–Meier curves were drawn. *p*-values were calculated with a log rank test.

## Data Availability

This study used TCGA RNA-seq dataset (https://portal.gdc.cancer.gov/repository (accessed on 1 June 2020)).

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
