# Peer review of "Pan-Cancer Profiling of Intron Retention and Its Clinical Significance in Diagnosis and Prognosis"

_cancers, 2023, doi:10.3390/cancers15235689_

Round 1
Reviewer 1 Report (Previous Reviewer 1)
Comments and Suggestions for Authors
The authors have thoroughly addressed my previous concerns, but please 1) revise Response 1 and incorporate it into Discussion and 2) provide supporting references that demonstrates the secretion of DIRs into the bloodstream and discuss this potential in the manuscript.
Author Response
|
1. Summary |
|
|
|
Thank you very much for taking the time to review our manuscript. We have performed necessary analyses and accordingly revised the manuscript. Please find the detailed responses below and the corresponding revisions (highlighted in blue fonts) in the re-submitted manuscript.
|
||
|
2. Questions for General Evaluation |
Reviewer’s Evaluation |
Response and Revisions |
|
Does the introduction provide sufficient background and include all relevant references? |
Must be improved |
|
|
Are all the cited references relevant to the research? |
Must be improved |
|
|
Is the research design appropriate? |
Yes |
|
|
Are the methods adequately described? |
Yes |
|
|
Are the results clearly presented? |
Yes |
|
|
Are the conclusions supported by the results? |
Yes |
|
|
3. Point-by-point response to Comments and Suggestions for Authors |
||
|
Comments 1: The authors have thoroughly addressed my previous concerns, but please 1) revise Response 1 and incorporate it into Discussion and 2) provide supporting references that demonstrates the secretion of DIRs into the bloodstream and discuss this potential in the manuscript.
|
||
|
Response 1: Thank you for your insightful considerations and kind advice. We have now incorporated Response 1 into Discussion (line #650 ~ #679). And we have also provided supporting references that demonstrates cell-free circulating tumor RNAs in plasma as the potential prognostic and diagnostic biomarkers, and discussed this potential for DIRs and prognostic IR events discovered in this study (line #628 ~ #641).
|
||
Reviewer 2 Report (Previous Reviewer 3)
Comments and Suggestions for Authors
The authors addressed most of my concerns and I find now the paper acceptable for publication, after addressing a couple of additional points:
1) I was really interested in the 7 additional oncogenes that were retained more in normal samples and spliced more in tumor samples - DDB2, MDM2, SETDB1, NKX2-1, PTK6, BCL2L12, CSF3R and RECQL4 - can the authors present analysis on what is the consequence of the IR for this genes, in a manner similar to STN1 (only the in silico predictions)? (my comment 4 and the response 4).
2) When the authors are discussing the sequence determinants for the DIR, I believe it is appropriate to cite the papers developing methods and analyzing the sequence determinants for DIR (PMID: 31779139, PMID: 30661751, PMID: 25481010).
Comments on the Quality of English LanguageThe quality of the language is good, may need some minor grammar and spelling checks.
Round 2
Reviewer 2 Report (Previous Reviewer 3)
Comments and Suggestions for Authors
The authors addressed my comments.
Comments on the Quality of English LanguageThe English is ok, just needs smoothing and polishing.
This manuscript is a resubmission of an earlier submission. The following is a list of the peer review reports and author responses from that submission.
Round 1
Reviewer 1 Report
Comments and Suggestions for Authors
Huang et al. identified the events of intron retentions (IRs), quantified their abundance, selected IRs that were differentially expressed between tumors and adjacent normals, revealed their potential of being diagnostic and prognostic biomarkers, and performed enrichment analyses using GO terms and KEGG pathways across 33 TCGA tumor types. In vitro assays were carried out in two immortalized cell lines, H1299 and A549, to help investigate whether an increased STN1 5’-UTR IR leads to decreased tumor colony formation and migration rates. Lastly, the authors regressed a LASSO model, IRR, using the selected IRs and found that patients with a lower IRR score have a prolonged survival rate across all tumor types tested in this study. This manuscript contains lots of fruitful information for studying the casual relationships between IRs and disease formation and progression. Their findings will be beneficial for researchers in the field and the future applications based on their results may have great clinical significance. Yet, a few issues still can be identified from the manuscript; listed below.
1) TCGA RNA-seq profiles were designed to capture mRNAs containing poly(A) tails; however, multiple studies have shown that intron reads of poly(A) sequencing data whose abundance were correlated well with the expression levels of nascent transcripts, i.e., pre-mRNAs (PMIDs: 33575621 & 26098447). In addition, many nuclear lncRNAs containing no poly(A) tail can still be identified from poly(A) sequencing data, suggesting that the sequencing library preparation is imprecise and prone to include non-poly(A) transcripts by chance. In this manuscript, the authors assumed 100% of intron reads were the products of intron retention. Please argue if this assumption is completely true and provide explanations why you and others have such disagreements.
2) The authors trained a Random Forest classifier based on the expressions of 273 DIRs and claimed that this model performed well in cancer type classification; however, I wonder if this model has any significance in clinical practice. During a surgical biopsy, surgeons were pretty much aware of what kind of tumor tissues being cut off. For example, you will never get breast tumors from a prostate. Instead, TCGA consortium have defined subtypes for each specific tumor type, and the authors should see if these DIRs are predictive of subtypes within the same tumor type. In addition, the authors should estimate the statistical significance of the AUCs in a non-parametric way. Namely, randomly selecting the same number (N=273) of genes/IRs 1K times to build Random Forest classifiers, then check how many of them have a better AUC than the one trained by the 273 DIRs.
3) While accessing the prognostic potential of IRs, the authors should have used the survival data corrected by Liu et al. (PMID: 29625055) to get the most accurate estimates.
4) The siRNA- or shRNA-mediated silencing mainly downregulates transcripts localized in cytoplasm. Please check if the relative transcript expressions of STN1 5’-UTR IR and STN1 spliced has been changed accordingly after shRNA silencing of STN1 (related to Figure 6E). Please provide evidence that the expression of STN1 5’-UTR IR was increased and this increase led to the phenotypic changes (related to Figure 6I-J). In addition, please consider overexpressing STN1 5’-UTR IR in the same cell line to see if the results in your Figure 6I-J will be phenocopied.
5) Minor:
a. Page 6: Please provide explanations why LAML, ESCA, and STAD had lower IR similarity with other tumors.
b. Page 7: What could be the reasons that LAML has more IR events than other tumor types? Are they simply artifacts?
c. Page 10: Please provide explanations why down- and up-regulated IRs both had a much shorter median length than unregulated ones.
d. Page 17: The legend is missing from Figure 7D.
Reviewer 2 Report
Comments and Suggestions for Authors
Minor:
1) Please change ‘bam’ to ‘BAM’ for refering to the Binary Alignment Map (BAM) format.
2) Line #112; Please change ‘Padjusted’ to ‘Adjusted p-value’.
3) Line #165; Please change ‘p value to ‘p-value’.
4) Figure 3D; Please add a dashed line to mark the mean of relative gene position.
5) Codes or workflows for the data analysis associated with the manuscript should be made available in the public repository.
Reviewer 3 Report
Comments and Suggestions for Authors
The presented work on pan-cancer profiling of intron retention is interesting, well written, with sound and well described methods. In general, I believe that this research deserves attention, after some important additions. My concerns and questions are below:
Major:
1) The authors state that AUC>0.9 for IR – is the DEG between cancer accounted for in this calculation?
2) It is not sufficiently clear– do the authors find more DIR in the oncogenes and TSGs?
3) I am intrigued by the lack of strong sequence determinants for the DIR. First, this section needs more comprehensive presentation (for example the last sentence needs to refer to concrete numbers. Second, and related to potential somatic variants, did the authors performed analyses for intron retention associated genetic variants (PMID: 31779139, PMID: 30661751, PMID: 25481010)?
4) How do the authors interpret events such as the one represented on Figure 4G? Intron retention, which is supposed to be aberrant compared to splicing, is seen more in normal (vs tumor) samples?
5) In the samples with the highest DIR, do the authors observe different levels of RNA for splice-relevant genes (i.e. RNA Pol II, genes from the spliceosome, etc?) based on the RNA-seq data?
Minor:
1) The five-pointed star on Fig 1 does not seem in place, I would suggest replacing with a different symbol.
2) Fig3 E-H – how was the tSNE applied? Are the dimensions the introns’ expression?
3) The term “not regulated” (for introns that are neither up- and down-regulated) is inaccurate.
Comments on the Quality of English LanguageLanguage check is needed. Also, as per the review, some terms need top be edited.
